# Habitual Total Drinking Fluid Intake Did Not Affect Plasma Hydration Biomarkers among Young Male Athletes in Beijing, China: A Cross-Sectional Study

**DOI:** 10.3390/nu14112311

**Published:** 2022-05-31

**Authors:** Jianfen Zhang, Na Zhang, Yibin Li, Hairong He, Guansheng Ma

**Affiliations:** 1Department of Nutrition and Food Hygiene, School of Public Health, Peking University, 38 Xue Yuan Road, Haidian District, Beijing 100191, China; zjf@bjmu.edu.cn (J.Z.); zhangna@bjmu.edu.cn (N.Z.); 1310306130@bjmu.edu.cn (Y.L.); hhrhhr3@163.com (H.H.); 2Laboratory of Toxicological Research and Risk Assessment for Food Safety, Peking University, 38 Xue Yuan Road, Haidian District, Beijing 100191, China

**Keywords:** hydration biomarkers, drinking patterns, total drinking fluids, water from food, young athletes

## Abstract

The purposes of this study were to explore the drinking patterns, and urinary and plasma hydration biomarkers of young adults with different levels of habitual total drinking fluid intake. A cross-sectional study was conducted among 111 young male athletes in Beijing, China. Total drinking fluids and water from food were assessed by a 7-day, 24-h fluid intake questionnaire and the duplicate portion method, respectively. The osmolality and electrolyte concentrations of the 24-h urine and fasting blood samples were tested. Differences in groups LD_1_ (low drinker), LD_2_, HD_1_, and HD_2_ (high drinker), divided according to the quartiles of total drinking fluids, were compared using one-way ANOVA, Kruskal–Wallis H-tests, and chi-squared tests. A total of 109 subjects completed the study. The HD_2_ group had greater amounts of TWI (total water intake) and higher and lower contributions of total drinking fluids and water from food to TWI, respectively, than the LD_1_, LD_2_, and HD_1_ groups (*p* < 0.05), but the amounts of water from food did not differ significantly among the four groups (all *p >* 0.05). Participants in the HD_2_ group had higher amounts of water than participants in the LD_1_, LD_2_, and HD_1_ groups (*p* < 0.05); SSBs were the second top contributor of total drinking fluids, ranging from 24.0% to 31.8%. The percentage of subjects in optimal hydration status increased from 11.8% in the LD_1_ group to 58.8% in the HD_2_ group (*p <* 0.05). The HD_2_ and HD_1_ groups had 212–227 higher volumes of urine than the LD_1_ and LD_2_ groups (*p* < 0.05). No significant differences were found in the plasma biomarkers (*p* > 0.05), with the exception of higher concentrations of K in the HD_1_ group than in the LD_1_ group (*p <* 0.05). Subjects with higher amounts of total drinking fluids had better hydration status than those with lower total drinking fluids, but not better drinking patterns. Habitual total drinking fluids did not affect the plasma biomarkers.

## 1. Background

Maintaining adequate water intake is crucial for human beings. If one’s water intake is less than the output of the water, the body may be in a state of dehydration. Research has demonstrated that dehydration can impede cardiovascular functions and even cognitive performance [1,2]. It takes more time to accumulate when the body mass has losses of 2% or more. Furthermore, if one’s water loss is more than 20% of their body weight, death occurs. Many factors influence the intake of total drinking fluids among individuals, including age, the temperature, the humidity of the environment, activity levels, and physical measurements [3,4,5]. Studies have shown that vigorous exercise increases the core temperature of the human body, and then the blood flow to the skin increases to expel the heat through sweat. Sweat loss during exercise that exceeds water intake may lead to dehydration, which increases the physiological stress and perception of effort, then reduces endurance performance [6]. Moreover, dehydration may lead to acute renal failure and heat illness among athletes. Consequently, it is important for athletes or people who carry out regular physical exercise to maintain adequate water intake.

Studies show that up to 50% of women and about 60% of men do not meet the recommendations for total drinking fluid intake of the European Food Safety Agency (EFSA) [7]. Furthermore, male adults had a higher risk of not complying with the reference value of total drinking fluids, and even among older people, 34% of the 156 people surveyed ingested less than 1500 mL/d f [8,9]. It has been shown that physical activities can affect fluid intake; moreover, during taking exercise, the lost both in water and electrolytes as a consequence of sweat. Athletes may have longer, high-intensity exercise than non-athletes who are less active, and the sweat loss rate, by weight, may be higher than their counterparts. Therefore, the intake of the athletes or people who carry out regular physical exercise may differ from their counterparts, and differences might even exist between athletes with different specialties. According to differences in the sweating rate, sweat sodium concentration, and sweat loss in athletes from different sports, athletes with football and endurance have the greatest demand for the deliberation of the hydration strategies [10]. Hence, it has been suggested that planned drinking, and not drinking to thirst, is optimal for athletes with longer duration activities of 90 min or more, especially in hot environments [11]. Therefore, it is important to investigate and elaborate on the characteristics of the drinking patterns among athletes. Nevertheless, research about the fluid intake among athletes in free-living conditions is scarce. One study demonstrated that the average fluid intake of 11 male and female elite ultra-endurance runners was 16.4 L [12], and that of eight female athletes was 19.18 L [13]. Furthermore, the amounts of the water intake could range from 3.2 L to 10.3 L, between rest and preseason, among male athletes [14]. In China, there were only two large studies that investigated fluid intake among adults and children [15,16], and only two studies explored water intake among young adults including males and females aged 18–23 years [17,18]; however, among athletes, no relevant studies have been conducted.

Studies have shown that the types of fluids consumed, including water and other beverages, were different among different age groups; milk consumption was higher in children, whereas tea, coffee, and alcoholic beverage consumption was a little higher among adults [19]. Moreover, among young adults in free-living conditions, subjects with high total drinking fluid intake had better drinking patterns, including higher water intake and lower intake of sugar-sweetened beverages (SSBs) than their counterparts with lower total drinking fluid intake [17]. For athletes, a small body of research exploring fluid consumption behavior suggests that different individuals may prefer consuming different types of fluids with different volumes after exercise. Furthermore, the types of fluids consumed could affect hydration status and muscle recovery performance among athletes [20,21]. Moreover, a randomized, controlled, double-blinded study demonstrated that electrolyte content seems to be the first contributor to the hydration indexes of beverages for young males and females when consumed at rest [22]. The type of fluids did, affect the nutrient intake among some athletes, during the post-exercise period. For example, beverages including sodium effectively reduced the urine volume and restored body water balance [23]. Furthermore, the types of fluids consumed may also influence fluid and electrolyte homeostasis in athletes before exercise [24]. To date, no related study exploring the types and amounts of fluids consumed has been carefully quantified in athletes in free-living conditions (they could do everything as usual, including learning, eating, sleeping and training).

Maintaining the adequate water intake or the optimal hydration status is crucial for the optimization of physical and cognitive performances. Moreover, it has been revealed that the comparing with the exogenous carbohydrate intake, the hydration status was more important for athletes doing push-to-the-finish cycle in the heat [25]. Despite the amount of water intake impacting hydration status, it is well known that the types of fluids consumed also affect hydration status. Among adults and children, those with higher total fluid intake had better hydration status than their counterparts with lower intake, both in France and China [17,26]. To assess the hydration status, there were many hydration biomarkers including the plasma and urinary biomarkers were used [27,28]. Comparing with plasma biomarkers including the plasma osmolality and the concentrations of the electrolytes, less invasive and alternative urine biomarkers were consisted of the osmolality, specific gravity (USG), volume and color. Furthermore, the urine osmolality was considered to be the “gold standard” method for evaluating hydration status. In the present study, the hydration status was assessed both with plasma and urinary biomarkers. Currently, it is unknown whether individuals who habitually consume high amounts of fluids are similar to those who drink lower volumes of fluids physiologically. Athletes lose a lot of water during vigorous exercise, and their drinking, urination, and hydration state may be different from those of the general population. Hence, the drinking patterns among people engaging in regular physical exercise with different total drinking fluid intake levels, and how they influence the hydration status, represent an interesting area of research.

The purposes of this study were, firstly, to explore the differences in drinking patterns among young athletes, and secondly, to investigate the hydration biomarkers of young adults with different levels of habitual total drinking fluid intake in free-living conditions. With this research, we aim to contribute to the provision of a science-based education on fluid intake for young athletes.

## 2. Methods

### 2.1. Subjects

A cross-sectional study was designed, and 111 young adult males were recruited in Beijing, China. Inclusion criteria were as follows: healthy adult male college students aged 18–25 years with a regular exercise training plan (more than 5 moderate-intensity exercises per week). Exclusion criteria were as follows: those with chronic diseases, such as oral cavity, endocrine, kidney, gastrointestinal tract, and metabolic diseases; sports injury; cognitive impairment; and those who have taken drugs or vitamins and other health products within 1 month.

### 2.2. Sample Size Calculation

In similar study, more than 75% of the random urine samples of 4.4% of the subjects were of optimal hydration status [18], with the set t = 1.96 (α = 0.05), e = 4%. Thus, according to the formula for calculating the sample size of simple random sampling, *n* = t^2^p (1 − p)/e^2^, the maximum sample size is 101. Considering the drop-out rate, set at 10%, 111 subjects were needed in this work.

### 2.3. Study Procedure

The study spanned 7 consecutive days, including five weekdays and two weekends. On the first study day, anthropometry including height, weight, and waist circumference, were measured. As for recording fluid intake from water and other beverages, all subjects were instructed several times to successfully record the related information on the self-designed 7-day, 24-h fluid intake questionnaire. In addition, over the 7 consecutive days, all the foods that the subjects ate were weighed and recorded for three consecutive days (two weekdays and one weekend day, from day 3 to day 5), as described in our previous study [17]. Furthermore, during the three days, 24-h urine samples, including the first morning urine, were collected by the subjects. On day 4, the fasting venous blood samples of all subjects were collected. Moreover, the indoor and outdoor temperature and humidity were recorded each day for 7 days. The study procedure is shown in Figure 1.

### 2.4. Measurement of Total Water Intake (TWI)

Total water intake included the total drinking fluids and water from food (mL).

A 7-day, 24-h fluid intake record questionnaire was used to assess the total drinking fluids, which was designed by the investigator, with high validity, as described before [15,16,17,18]. Subjects were asked to accomplish the questionnaire for 7 consecutive days (five weekdays and two weekends). The type and amount of fluid intake for each time were measured by a standard cup, which was provided by the investigator. Total drinking fluids were grouped into 6 categories according to the General Standard for Beverages of China (GB/T 1-789-2015) [29], as described before [30]. Furthermore, to ensure completeness and accuracy, the investigators checked the questionnaire every day.

All foods that the subjects ate for the three days were weighed before and after subjects ate to calculate the amount of water from food. Water from food was assessed with the duplicate portion method. Moreover, all the samples of food were collected by the investigators and sent to the laboratory to be stored immediately. Samples of all foods were measured according to the national standard of GB 5009.3-2016 [31], and the water from fruits or other snacks was assessed according to the China Food Composition Table (2009) [32]. The water from food was separated into five categories, as described in our previous study [30].

### 2.5. Temperature and Humidity of the Environment

The indoor and outdoor temperature and humidity were recorded each day for 7 days (WSB-1-H2, Exasace, Zhengzhou, China). The outdoor temperature and humidity were 24.2 °C ± 5.8 °C and 29.5% ± 15.8%.

### 2.6. Anthropometric Measurements

Height, weight, and waist circumference were measured twice by the trained investigators with standard procedure [33,34]. The participants were asked to wear light clothes and no shoes (BSM370; BIOSPACE; Seoul, Korea). BMI: weight (kg)/height squared (m^2^). 

### 2.7. Urine Biomarkers

The 24-h urine samples, including the first morning urine, were collected by the subjects using self-designed containers by the investigators. Each urine sample was asked to be collected by the participants and sent to the investigators, and then the investigators brought the samples of urine to the laboratory to be stored. All the urine samples were stored at +4 °C in a special refrigerator for samples before being measured. When the urine samples were sent to the laboratory, the investigators recorded the weight of the samples using the desktop electronic scale (YP20001, SPC, Shanghai, China), to the nearest 0.1 kg. Urine osmolality, urine specific gravity (USG), pH, urea, and creatinine, urine electrolyte concentrations (including sodium, potassium, chloride, calcium, magnesium, and phosphate) were tested with standard procedure, which was described in our previous study [17]. Optimal hydration is defined when urine osmolality ≤ 500 mOsm/kg; medium hydration is defined as 500 mOsm/kg < urine osmolality ≤ 800 mOsm/kg; and dehydration is defined as urine osmolality > 800 mOsm/kg [18].

### 2.8. Blood Biomarkers

Fasting venous blood samples were collected with standard procedure by trained investigators to measure the concentrations of Na, K, and Cl, and testosterone, cortisol, creatinine, and copeptin.

The concentrations of electrolyte (including the sodium, potassium, chloride, calcium, magnesium, and phosphate) were tested, as described before in our previous study [14]. The testosterone, cortisol, and copeptin levels were determined by a trained investigator using Imark microplate reader (Bio-Rad 680, Bio-Rad, Hercules, CA, USA). The creatinine level was assessed using the sarcosine oxidase method (C011-2-1, Jiancheng, Nanjing, China). 

### 2.9. Physical Activity

Participants were asked to record the information of the physical activity during the 7 days on the questionnaire, which was self-designed by the researchers, including the time, the type, and the place of the activities. Furthermore, the training program of the participants was provided to the researchers by the coaches that guided them. The types of the sports were aerobics, table tennis, tennis, football, track-and-field, and others. 

### 2.10. Statistics

The SAS 9.2 software (SAS Institute Inc., Cary, NC, USA) was used for statistical analysis. Data were presented as mean ± standard deviation, and median or interquartile ranges if the data were normally distributed or not, respectively. Subjects were divided into four groups, LD_1_ (low drinker 1), LD_2_ (low drinker 2), HD_1_ (high drinker 1), and HD_2_ (high drinker 2), according to the quartiles of total drinking fluids of the subjects (Q_1_: 947–1422 mL, Q_2_: 1460–1789 mL, Q_3_: 1798–2297 mL, Q_4_: 2311–3652 mL). One-way ANOVA, Kruskal–Wallis H-tests, and chi-squared tests were used to compare the differences among the four groups. Differences between each two groups were compared using SNK (*p* < 0.05). The significance level was set at 0.05 (*p* < 0.05). 

## 3. Results

In total, 111 subjects were recruited, and 109 of them completed the study, giving a completion rate of 98.2%. Table 1 shows the characteristics of the participants. However, the age, height, weight, BMI, and skeletal muscle did not differ significantly among the four groups (all *p* > 0.05).

### 3.1. Measurement of TWI

TWI increased simultaneously, with the intake of the total drinking fluids increased (*p* < 0.05). Subjects in HD_2_ had 1433–1620 mL, 1170–1203 mL, and 733–853 mL more than other athletes in groups LD_1_–HD_1_ in the amounts of total drinking fluids and TWI (*p* < 0.05), respectively. There were no significant differences in the amounts of water from food among the four groups (*p* > 0.05). According to the proportions of total drinking fluids in TWI, they were different among the four groups compared with each other (*p* < 0.05), which ranged from 56.9% in group LD_1_ to 72.7% in group HD_2_. Refer to the water from food, accounting for 27.3–43.1% of TWI from group LD_1_ to group HD_2_, these levels also differed significantly among the four groups (*p* < 0.05). Subjects in group HD_2_ had a 20.4% higher and 20.4% lower contribution of total drinking fluids and water from food to TWI than those in group LD_1_, respectively (*p* < 0.05).

Regarding the drinking patterns, which included the amount of fluids intake and the types and contributions of fluids, the main contributor to total drinking fluids was water in all the four groups, accounting for 61.2–69.7%, and there were no significant differences among the groups (*p* > 0.05). The amounts of water differed significantly among the four groups (*p* < 0.05), and increased with higher total drinking fluids. However, no statistically significantly differences were found in the consumptions of beverages including the tea, milk and milk products, or alcoholic beverages among the four groups (all *p* > 0.05), except the amounts of SSBs and other beverages (all *p* < 0.05). For SSBs, the consumption in the LD_1_ group was lower than in other groups (*p* < 0.05). Furthermore, the contributions of water, tea, milk and milk products, SSBs, or alcoholic beverages to total drinking fluids did not differ significantly among the four groups (all *p* > 0.05), except the contributions of other beverages (*p* < 0.05). Surprisingly, the SSBs were the second top contributor to total drinking fluids after water in all the groups, ranging from 24.0% to 31.8%. 

Regarding the water from food, subjects in HD groups had similar water from food intake with those in LD groups. In all four groups, the main source of water from food was dishes, with staple food following by. However, there were no statistically significant differences in the amounts of water from staple food, dishes, soup, porridge, and snacks in the four groups (all *p* > 0.05), neither in the contributions of water from different foods, nor in the contribution of staple food, dishes, soup, porridge, and snacks (all *p* > 0.05), as shown in Table 2.

### 3.2. Measurement of Urine Indexes

Table 3 revealed that, with the increase in the total drinking fluids, the volumes of urine also increased from groups LD_1_ to HD_2_, while the osmolality and the concentrations of Na and Cl were different in LD_1_, LD_2_, HD_1_, and HD_2_ (*p* < 0.05). The proportions of subjects with optimal hydration status were higher in HD_2_ and HD_1_ than that in LD_1_ and LD_2_ (*p* < 0.05).

### 3.3. Measurement of Blood Indexes

No statistically significant differences were found in the concentrations of Na, Cl, copeptin, testosterone, cortisol, and creatinine among subjects in the four groups (*p* > 0.05), except the concentration of K, as shown in Table 4.

## 4. Discussion

The present study is the first to investigate the differences among athletes with different total drinking fluid in China. The results of the present study show that the consumptions of total drinking fluids, TWI, and water from food of subjects were with 447–654 mL, 148–359 mL more, and 219–256 mL less than the results of survey conducted among young adults in China, respectively [17]. This could be because the participants in the current study were active athletes compared to the young adults in other studies. Similarly, the results of a study conducted in Japan and in European countries on different physical activity levels of people confirmed with ours. For children, the results of a study including 7229 children from 8 countries revealed that children including the boys and girls, who spending less time performing moderate to vigorous physical activity, consumed less water than those with high levels of moderate to vigorous physical activity [35].

In the current study, among the four groups, when the total drinking fluids increased significantly, the TWI also increased significantly, and the percentages of total drinking fluids in TWI were increased. Simultaneously, the contributions of water from food to TWI were decreased, from 47.1% to 27.3%, much lower than reported in studies on young adults in Baoding, China, and adults from four cities of China [17,36]. Surprisingly, the intakes of water from food were largely similar among the four groups and consistent with the results of a study performed among Europeans [26], but different from the results of a study in China [17]. The findings indicate that subjects with lower total drinking fluid intake would not compensate with water from food to increase the intake of TWI. We can conclude that the main differences in the TWI among the young athletes were from the total drinking fluids, need to pay more attention to daily fluid demands. The interventions to increase the TWI of young adults should focus on the improvement of total drinking fluids. For the types of fluid intake, water was the major source in the four groups, and subjects with higher total fluid intake were more likely to consume more water. Nevertheless, the present study showed that the second main contributors to total drinking fluids followed water were SSBs. According to the WHO, the consumption of SSBs is a harmful dietary habit and is considered to be a risk factor for chronic non-communicable diseases (NCD) because it leads to an increase in free sugar intake. It has been shown that higher consumptions of SSBs are related to higher hazard of hypertension, type 2 diabetes, excessive daytime sleepiness, and obesity-related cancers [37,38,39,40]. SSBs may include sports drinks or energy drinks for the young adults in our study, but studies concluded that even among the athletes, the impacts of ingesting high amounts of SSBs on the health of metabolism, the levels of blood glucose and insulin, should be taken into consideration [41]. More recently, approximately 96% of caffeine consumption from beverages comes from coffee, soft drinks, and tea. Even the caffeine could improve the physical performances among athletes, the adverse effects including impacted the sleep or increased the feelings of anxiety should be paid attention. Hence, interventions to reduce the intake of SSBs and develop methods to select appropriate sports drinks should be proposed.

The results of the study demonstrated several differences in urinary biomarkers between participants with HD_2_ and LD_1_. In both the volume and osmolality, the LD_1_ group had statistically significant lower urine volume coupled with higher osmolality than the HD_2_ group, similar to the results of studies in China [17] and France [26]. Furthermore, the hydration status was better in the HD_2_ group than the LD_1_ group. This indicates that subjects with higher total fluid intake had better hydration status than those with lower total drinking fluid intake. The proportion of subjects with optimal hydration status was only 15.6%, which is much lower than that of young male and female adults in China [42], revealing the athletes are more likely to be dehydrated. Similarly, in one study conducted on schoolchildren with physical activity and sedentary behavior in Spain, it was revealed that children who were more inactive were adequately hydrated [43]. Recent data reported that athletes are subjected to the adverse effects of dehydration. A study conducted on 28 male collegiate soccer players revealed that USG was greater than 1.020 on 12/15 days, indicating that participants experienced dehydration 80% of the time [44]. This highlights the importance of raising awareness about hydration status among young athletes. Furthermore, 63 youth football players with physical activity were mildly hypohydrated in free-living conditions [45]. Research has demonstrated that dehydration impedes physical performance. Studies have shown that personalized hydration strategies have an important role in majorizing the performance and safety of athletes in sporting activities [46]. Hence, it is important to propose intervention strategies to improve the hydration status of young athletes in China to mitigate the adverse effects of dehydration. Furthermore, health education about adequate water intake should also be proposed.

Hormones potentially impact the performance and the status of health among athletes [47]. According to the plasma biomarkers, it was somewhat counterintuitive that similar levels of copeptin, cortisol, creatinine, and testosterone were observed among subjects in the four groups in our study. Contrarily, another study demonstrated that the maintenance of the balance of fluid was modulated by hormones in young women with different levels of TWI with AVP, and not aldosterone, which was mainly responsible for the maintenance of body water and tonicity over days in participants with high and low water intake [48]. Moreover, the concentrations of cortisol, creatinine, and AVP in plasma were all higher in those with 1.2 L/day of total drinking fluids in the LD group than those with 2.0 L/day in the HD group among adults in France [26]. 

As is known, copeptin is associated with type 2 diabetes and heart diseases [49]. The present results indicate physiological adaptions to preserve the plasma biomarkers despite the broader amounts of total drinking fluids. A previous study showed that urine biomarkers, including the osmolality, USG, and color, responded to the changes in water intake, but not the plasma biomarkers [50]. The present study confirmed these conclusions. 

The circulating concentration of cortisol is monitored by the hypothalamus pituitary adrenal axis and the neuroendocrine feedback circuit, which can be activated by physiological stimuli including stress and exercise. The increase of the serum cortisol has been linked with hypohydration in some strenuous conditions. In our study, no significant differences in the plasma cortisol were found between subjects with different levels of total drinking fluid intake, which is similar with the results of other studies. The research supported the results showed that cortisol levels remained unchanged after exercise-induced dehydration, resulting in an average weight loss of 1.3% to 1.6% [51]. Another study found no adverse effects of dehydration induced by the combination of heat exposure and fluid restriction on cortisol [52]. However, these findings were not consistent with other studies; one showed that cortisol was significantly elevated with dehydration among adults, with a 2.72 nmol/L increase in cortisol for each 1% increase in total body mass loss [53]. Furthermore, studies evaluating the salivary cortisol demonstrated that salivary cortisol increased after exercise, but with no change in testosterone [54,55]. 

To the best of our knowledge, this is the first study to document the concentrations in plasma cortisol among athletes in free-living conditions even without acute dehydration. Our results confirmed the previous findings, expanding the knowledge for young adults regarding the effects of different levels of total drinking fluids among athletes with free-living conditions. The proportion of subjects with higher total drinking fluids was 40% higher than participants with lower total drinking fluids. Moreover, these data suggest that there is a threshold value of different levels of total drinking fluids on plasma cortisol among young athletes. 

The study has some strengths and weakness. As for the strengths, firstly, the total drinking fluid intake was evaluated by a 7-day, 24-hour fluid questionnaire, which is an effective and reliable method of estimating total drinking fluids in other studies before [15]. Moreover, subjects were asked to record the information including the time, volume, type for each fluid they drank, which provided more information to assess the drinking patterns. Secondly, before and after the subjects ate, all the foods were weighed, furthermore, the samples of food were measured with standard procedure by trained investigators. However, female athletes were not included in the present study. Furthermore, the whole body sweating rate [56] was not explored in our study. Therefore, in the future, studies with large sample sizes and the measurement of the body sweating rate are needed. 

## 5. Conclusions

Subjects with higher total drinking fluid intake had better hydration status than those with lower, but not better drinking patterns, even though the former had higher TWI than the latter. Furthermore, habitual total drinking fluids did not affect the plasma biomarkers. Interventions should be taken to encourage athletes to consume a sufficient amount of total drinking fluids and a healthier choice of beverages, in order to maintain optimal hydration.

## Figures and Tables

**Figure 1 nutrients-14-02311-f001:**
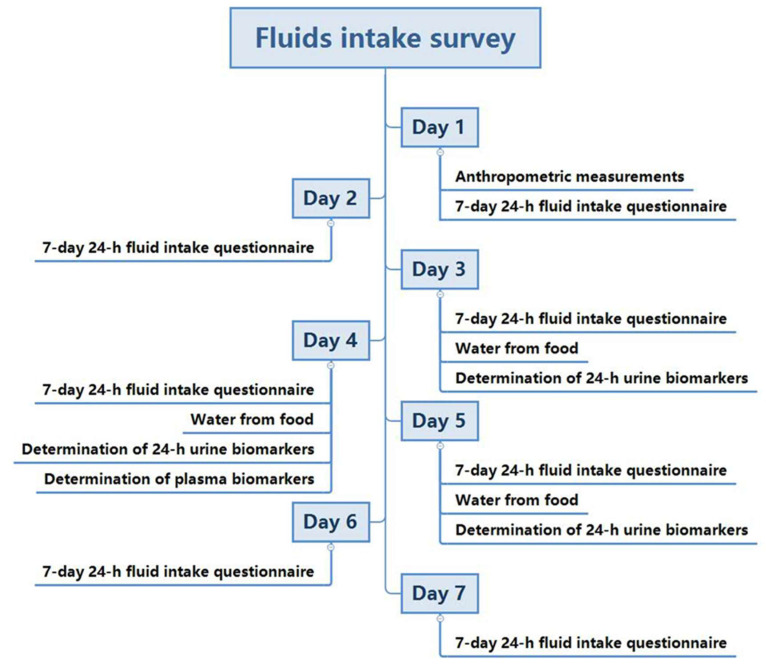
The study procedure.

**Table 1 nutrients-14-02311-t001:** The characteristics of participants.

	LD_1_ (*n* = 27)	LD_2_ (*n* = 28)	HD_1_ (*n* = 27)	HD_2_ (*n* = 27)	Total (*n* = 109)	*p*
Age (year)	20.4 ± 1.1	20.6 ± 1.0	20.9 ± 1.2	20.8 ± 1.0	20.8 ± 1.0	0.308
Height (cm)	177.4 ± 4.1	178.4 ± 5.1	178.1 ± 5.2	180.8 ± 5.8	178.7 ± 5.2	0.082
Weight (kg)	71.2 ± 5.9	69.9 ± 6.4	69.5 ± 8.1	72.2 ± 8.6	70.7 ± 7.3	0.500
BMI (Kg/m^2^)	22.6 ± 1.5	22.0 ± 2.1	21.9 ± 2.2	22.1 ± 2.1	22.1 ± 2.0	0.545
Skeletal muscle	34.1 ± 2.5	34.7 ± 2.7	34.6 ± 3.2	36.2 ± 4.2	34.9 ± 3.3	0.116

Note: Values are shown as the mean ± standard deviation (SD). BMI: Body Mass Index.

**Table 2 nutrients-14-02311-t002:** The TWI, total drinking fluids, and water from food among participants consuming different levels of total drinking fluids.

	LD_1_ (*n* = 27)			LD_2_ (*n* = 28)			HD_1_ (*n* = 27)			HD_2_ (*n* = 27)			Total (*n* = 109)		
	M	Q	%	M	Q	%	M	Q	%	M	Q	%	M	Q	%
Total drinking fluids	1252 ^a^	280	56.9 ^a^	1578 ^d,e^	140	63.3 ^d,e^	1952 ^b^	280	67.2 ^b^	2685 ^c,f^	687	72.7 ^c,f^	1789	863	65.0
Water	720 ^a^	414	64.0	951 ^d,e^	349	61.2	1370 ^b^	255	64.4	1934 ^c,f^	653	69.7	1181	666	64.8
Tea	0	0	1.0	0	0	0.5	0	0	0.7	0	0	0.3	0	0	0.6
Milk and milk products	31	107	7.4	33	104	3.8	50	99	4.4	50	196	4.3	40	111	5.0
SSBs	231	241	25.4	554	334	31.8	535	323	27.5	605 ^c^	422	24.0	469	424	27.2
Sports drinks	0	71	3.3	71	139	5.4	71	169	6.1	141 ^c^	193	6.8	65	154	5.4
Other SSBs	231	241	22.0	440	275	26.4	429	282	21.4	423 ^c^	427	17.2	383	365	21.8
Alcohol	0	0	2.2	0	45	2.7	0	86	3.0	0	0	1.2	0	0	2.3
Others	0	0	0.0	0 ^e^	0	0.0	0	0	0.1	0 ^c,f^	0	0.5	0	0	0.1
Water from food	894	363	43.1 ^a^	946	381	36.7 ^d,e^	1002	579	32.8 ^b^	960	421	27.3 ^c,f^	955	472	35.0
Staple food	330	213	39.4	322	92	35.0	324	99	35.6	363	113	36.2	330	107	36.5
Dishes	458	224	44.9	479	222	48.0	407	213	47.7	473	225	46.2	458	213	46.7
Soup	0	0	2.0	0	31	3.5	0	85	3.7	0	0	3.7	0	0	3.3
Porridge	56	179	10.3	48	130	8.5	43	146	8.1	83	171	11.1	60	151	9.5
Snacks	0	56	3.3	14	64	4.9	17	69	4.9	0	72	2.8	9	64	4.0
Total water intake	2133 ^a^	569	_	2550 ^d^	333	_	2900 ^b,f^	712	_	3753 ^c,e^	864	_	2701	973	_

Note: Values are shown as the median (M) and quartile ranges (Q); a: There was a statistically significant difference between LD_1_ and LD_2_ groups, *p* < 0.05; b: There was a statistically significant difference between LD_1_ and HD_1_ groups, *p* < 0.05; c: There was a statistically significant difference between LD_1_ and HD_2_ groups, *p* < 0.05; d: There was a statistically significant difference between LD_2_ and HD_1_ groups, *p* < 0.05; e: There was a statistically significant difference between LD_2_ and HD_2_ groups, *p* < 0.05; f: There was a statistically significant difference between HD_1_ and HD_2_ groups, *p* < 0.05. %: Contributions of total drinking fluids and water from food to TWI; percentages of different fluids in total drinking fluids; proportions of water from different foods in water from food. There were statistical significances in the amounts of TWI and total drinking fluids (*χ*^2^ = 77.958, *p* < 0.001; *χ*^2^ = 101.255, *p* < 0.001;) among the four groups, respectively, but no significant differences in the amounts of water from food (*χ*^2^= 1.158, *p* = 0.763). There were statistical significances in the consumption of water, SSBs, sports drinks, other SSBs, and others among the four groups (*χ*^2^ = 68.000, *p* < 0.001; *χ*^2^= 18.314, *p* < 0.001; *χ*^2^= 15.260, *p* = 0.002; *χ*^2^ = 11.434, *p* = 0.010; *χ*^2^ = 9.148, *p* = 0.027), with no significant differences were found in the volumes of milk and milk products, tea, and alcohol (*χ*^2^= 2.802, *p* = 0.423; *χ*^2^= 1.473, *p* = 0.689; *χ*^2^= 2.939, *p* = 0.401). There were statistical significances in the contributions of total drinking fluids and water from food to TWI, respectively (*F* = 23.096, *p* < 0.001; *F* = 23.096, *p* < 0.001). The contributions of water, milk and milk products, tea, alcohol, other SSBs, sports drinks, and SSBs in total drinking fluids did not differ significantly among the four groups (*F* = 1.144, *p* = 0.335; *F* = 1.561, *p* = 0.203; *F* = 0.354, *p* = 0.786; *F* = 0.582, *p* = 0.628; *F* = 2.394, *p* = 0.073; *F* = 0.968, *p* = 0.411; *F* = 1.355, *p* = 0.261), and significant differences were found in the contributions of others (*F* = 2.912, *p* = 0.038). No significant differences were found in the volumes of water from staple food, dishes, soup, porridge, and snacks (*χ*^2^= 2.709, *p* = 0.439; *χ*^2^ = 1.406, *p* = 0.704; *χ*^2^ = 0.420, *p* = 0.936; *χ*^2^ = 1.314, *p* = 0.726; *χ*^2^ = 2.267, *p* = 0.519). No significant differences were found in the contributions of staple food, dishes, soup, porridge, and snacks to water from food (*F* = 1.408, *p* = 0.245; *F* = 0.531, *p* = 0.662; *F* = 0.363, *p* = 0.780; *F* = 0.484, *p* = 0.694; *F* = 0.656, *p* = 0.581).

**Table 3 nutrients-14-02311-t003:** The characteristics of 24-h urine among participants consuming different levels of total drinking fluids.

	LD_1_ (*n* = 27)		LD_2_ (*n* = 28)		HD_1_ (*n* = 27)		HD_2_ (*n* = 27)		Total (*n* = 109)	
	M	Q	M	Q	M	Q	M	Q	M	Q
Volume (mL)	711	386	726	534	876	281	938	513	850	408
Urine Osmolality (mOsm/kg)	858	292	774	200	732	303	631	379	764	286
(≤500 mOsm/kg, *n*, %)	2 (11.8%) ^a,^*		1 (5.9%) ^a,b^		4 (23.5%) ^b^		10 (58.8%) ^b^		17 (15.6%)	
Void	3.8 ± 1.2		3.8 ± 1.3		3.9 ± 1.1		4.2 ± 1.3		3.9 ± 1.2	
Na (mmol/L)	229	63	212	52	186	74	192	85	202	66
K (mmol/L)	45.45	14.59	48.25	9.76	42.14	13.99	41.29	15.00	45.21	12.93
Cl (mmol/L)	222	35	229	35	203	49	218	83	221	53
USG	1.020	0.010	1.022	0.007	1.020	0.007	1.020	0.008	1.020	0.007
pH	6.7	0.7	6.2	0.6	6.3	0.5	6.3	0.6	6.3	0.5

Note: Values are shown as the median (M) and quartile ranges (Q); * *χ*^2^ = 20.274, *p* = 0.002; a: There was a statistically significant difference between LD_1_ and LD_2_ groups, *p* < 0.05; b: There was a statistically significant difference between LD_1_ and HD_1_ groups, *p* < 0.05. Significant differences were found in the volume, osmolality, and the concentrations of Na and Cl (*χ*^2^ = 9.141, *p* = 0.027; *χ*^2^ = 12.831, *p* = 0.005; *χ*^2^ = 9.900, *p* = 0.019), but no significant differences were found in the voids, K, USG, and pH among the four groups (*F* = 0.567, *p* = 0.638; *χ*^2^ = 6.627, *p* = 0.085; *χ*^2^ = 2.945, *p* = 0.400; *χ*^2^ = 6.148, *p* = 0.105).

**Table 4 nutrients-14-02311-t004:** The characteristics of blood samples among participants consuming different levels of total drinking fluids.

	LD_1_ (*n* = 27)	LD_2_ (*n* = 28)	HD_1_ (*n* = 27)	HD_2_ (*n* = 27)	Total (*n* = 109)	*p*
Copeptin (ρmmol/L)	1.68 ± 0.14	1.66 ± 0.15	1.65 ± 0.12	1.68 ± 0.10	1.67 ± 0.13	0.776
Testosterone (nmol/L)	17.4 ± 2.6	17.5 ± 2.6	16.4 ± 2.2	16.4 ± 2.2	16.9 ± 2.5	0.177
Cortisol (ng/L)	80.9 ± 11.4	81.4 ± 16.9	84.1 ± 14.9	96.9 ± 15.7	83.3 ± 14.9	0.425
Creatinine (μmmol/L)	65.5 ± 14.2	65.2 ± 16.8	65.5 ± 15.0	66.8 ± 14.1	65.7 ± 14.9	0.980
Na (mmol/L)	141 ± 3	142 ± 3	141 ± 3	140 ± 4	141 ± 4	0.327
K (mmol/L)	4.06 ± 0.51	4.20 ± 0.64	4.31 ± 0.63	4.63 ± 0.73 ^c^	4.30 ± 0.66	0.010
Cl (mmol/L)	104 ± 8	102 ± 6	104 ± 7	102 ± 7	103 ± 7	0.534

Note: Values are shown as the mean ± standard deviation (SD); c: There was a statistically significant difference between LD_1_ and HD_2_ groups, *p* < 0.05. No significant differences were found in the concentrations of copeptin, testosterone, cortisol, creatinine, Na, and Cl among the four groups (*F* = 0.368, *p* = 0.776; *F* = 1.676, *p* = 0.177; *F* = 0.938, *p* = 0.425; *F* = 0.061, *p* = 0.980; *F* = 0.734, *p* = 0.534; *F* = 1.163, *p* = 0.327), except the concentrations of K (*F* = 3.965, *p* = 0.010).

## Data Availability

The datasets generated and/or analyzed during the current study are available from the corresponding author on reasonable request.

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
