# Peer review of "Habitual Total Drinking Fluid Intake Did Not Affect Plasma Hydration Biomarkers among Young Male Athletes in Beijing, China: A Cross-Sectional Study"

_nutrients, 2022, doi:10.3390/nu14112311_

Round 1

Reviewer 1 Report

I have minor suggestions for the authors.

  1. Is not clear if the drinking patterns is measured as TWI. If so, I think that TWI is not stragly related to pattern. Suggest to clarify.
  2. If principal variable is drinking pattern why authors consider hydration status to estimate the samplea size?

Author Response

Authors' Response to Reviewers' Comments Reviewer 1 Open Review (x) I would not like to sign my review report ( ) I would like to sign my review report English language and style ( ) Extensive editing of English language and style required ( ) Moderate English changes required ( ) English language and style are fine/minor spell check required (x) I don't feel qualified to judge about the English language and style Yes Can be improved Must be improved Not applicable Does the introduction provide sufficient background and include all relevant references? ( ) (x) ( ) ( ) Is the research design appropriate? (x) ( ) ( ) ( ) Are the methods adequately described? ( ) (x) ( ) ( ) Are the results clearly presented? ( ) (x) ( ) ( ) Are the conclusions supported by the results? (x) ( ) ( ) ( ) Comments and Suggestions for Authors I have minor suggestions for the authors. 1. Is not clear if the drinking patterns is measured as TWI. If so, I think that TWI is not strongly related to pattern. Suggest to clarify. Response: Thanks for your comments. It was known that the TWI was consisted of the total drinking fluids and the water from food. Furthermore, studies have shown that the TWI and total dinking fluids were associated with the urinary hydration biomarkers, including the osmolality, the volume and the concentrations of the electrolytes. It has been concluded that the amounts and types of fluids consumed affect hydration status. In our study, the drinking patterns were the amounts, the types of total drinking fluids, which were assessed by a 7-day, 24-hour fluid intake questionnaire. Meanwhile, the information of the water from food of the young athletes were collected by the duplicate portion method for 3 days, including two weekdays and one weekend. Moreover, the types of fluids consumed could not only affect hydration status but also the muscle recovery performance among athletes. Therefore, it was of vital importance to investigate the drinking patterns among young athletes. 2. If principal variable is drinking pattern why authors consider hydration status to estimate the sample size? Response: Thanks for your comments. The study was one of the Project of “Survey of the drinking behavior of the young athletes”. The aims of the Project were to explore the characteristics of the total drinking fluids behavior, the urinary behavior, to investigate the relationship between the water intake and hydration status, and to find out the differences in different activities among young male athletes in Beijing. The purposes of this study were, firstly, to explore the differences in drinking patterns among young adults, and secondly, to investigate the hydration biomarkers of young adults with different levels of habitual total drinking fluid intake in free-living conditions. Furthermore, we also aimed to contribute to the provision of a science-based education on fluid intake for young adults. It has been showed that 37.2%~37.5% of the young adults were in dehydration, assessed by the osmolality of urine, among the young adults in China. Furthermore, there were about 18.6%~23.5% of the young adults met the recommendation of total drinking fluids of China. However, there were no studies investigating the related information including the amounts, types of drinking fluids of young athletes in China. Therefore, it was important to investigate the drinking patterns of the young athletes. We calculated the sample size according to the studies of young adults using the drinking patterns and the hydration status, the sample sizes were 98 and 111, respectively. Therefore, the final sample size of this study was 111, which was calculate by the hydration status. References: [1] Zhang N, Du S, Tang Z, et al. Hydration, Fluid Intake, and Related Urine Biomarkers among Male College Students in Cangzhou, China: A Cross-Sectional Study-Applications for Assessing Fluid Intake and Adequate Water Intake. International Journal of Environmental Research and Public Health, 2017, 14(5). [2] Zhang J, Zhang N, Wang Y, et al. Drinking patterns and hydration biomarkers among young adults with different levels of habitual total drinking fluids intake in Baoding, Hebei Province, China: a cross-sectional study. BMC Public Health, 2020, 20:468. [3] Zhang J, Zhang N, Liang S, et al. The amounts and contributions of total drinking fluids and water from food to total water intake of young adults in Baoding, China. European journal of nutrition, 2018, 58:2669-2677.

Reviewer 2 Report

In the abstract in the abbreviation TWI it is pending to indicate its meaning. This section should be explained by itself, without the need to go into the full article. The recommendations for change indicated in the complete article should be applied.

In the introduction section

In the first paragraph there is a sentence without a bibliography.

In the second paragraph, in line 64, I understand that it refers to reference 12, not to 11. In this same paragraph, in line 61, it indicates a study in an intake in liters and another in milliliters, it is recommended to use in both quantities the same unit of measure.

In the third paragraph the authors indicate the term free life, what do they mean by this term? It is recommended that the terms used are in the MeSH database (https://www.ncbi.nlm.nih.gov/mesh/) or that reviews and/or meta-analyses of this area of ​​knowledge have been previously published.

Several points of interest could be distinguished in this area of ​​knowledge: i. Water and hydration in sports, with the intake of carbohydrates and micronutrients to improve sports performance; ii. Methods for analyzing body weight loss after training or competition; iii. Changes in biochemical markers as an effect of hydration or dehydration. Despite these points described in the introduction, the objectives set are not justified. It is recommended to focus on the objective of analyzing the relationship in hydration in young athletes, for that perhaps the following bibliography can help:

  • Armstrong LE. Rehydration during Endurance Exercise: Challenges, Research, Options, Methods. Nutrients. 2021 Mar 9;13(3):887. doi: 10.3390/nu13030887
  • Baker LB, Ungaro CT, Sopeña BC, Nuccio RP, Reimel AJ, Carter JM, Stofan JR, Barnes KA. Body map of regional vs. whole body sweating rate and sweat electrolyte concentrations in men and women during moderate exercise-heat stress. J Appl Physiol (1985). 2018 May 1;124(5):1304-1318. doi: 10.1152/japplphysiol.00867.2017
  • Baker LB. Sweating Rate and Sweat Sodium Concentration in Athletes: A Review of Methodology and Intra/Interindividual Variability. Sports Med. 2017 Mar;47(Suppl 1):111-128. doi: 10.1007/s40279-017-0691-5
  • Barnes KA, Anderson ML, Stofan JR, Dalrymple KJ, Reimel AJ, Roberts TJ, Randell RK, Ungaro CT, Baker LB. Normative data for sweating rate, sweat sodium concentration, and sweat sodium loss in athletes: An update and analysis by sport. J Sports Sci. 2019 Oct;37(20):2356-2366. doi: 10.1080/02640414.2019.1633159
  • Campbell B, Wilborn C, La Bounty P, Taylor L, Nelson MT, Greenwood M, Ziegenfuss TN, Lopez HL, Hoffman JR, Stout JR, Schmitz S, Collins R, Kalman DS, Antonio J, Kreider RB. International Society of Sports Nutrition position stand: energy drinks. J Int Soc Sports Nutr. 2013 Jan 3;10(1):1. doi: 10.1186/1550-2783-10-1
  • Kenefick RW. Drinking Strategies: Planned Drinking Versus Drinking to Thirst. Sports Med. 2018 Mar;48(Suppl 1):31-37. doi: 10.1007/s40279-017-0844-6

If the target subjects are young adult athletes, it is recommended to delete all or most of the information on older adults and non-athletic adults. In addition, the authors must decide if this study is focused on improving the performance or health status of athletes.

The objective of this study may be to analyze whether athletes apply good hydration guidelines. And in the event that they do not have good hydration guidelines, then in the discussion the recommendations that are necessary in the case of future studies will be indicated. Therefore, the comments on line 97 are not appropriate for this section.

In the methodology section

The authors must declare whether they have followed the recommendations of the Declaration of Helsinki (World Medical Association Declaration of Helsinki: ethical principles for medical research involving human subjects. JAMA. 2013 Nov 27;310(20):2191-4. doi: 10.1001/ jama.2013.281053) and inform if the study has been approved by an ethics committee.

The type of sports practiced by the study subjects should be indicated and the scale used to indicate that the athletes performed moderate-intensity exercise should be indicated, as recommended by the scale proposed by the American College of Sports Medicine (Quantity and quality of exercise for developing and maintaining cardiorespiratory, musculoskeletal, and neuromotor fitness in apparently healthy adults: guidance for prescribing exercise. Med Sci Sports Exerc. 2011;43(7):1334-59.doi:10.1249/MSS.0b013e318213fefb).

In the measurement of Total Water Intake, why are the methods described appropriate and not others?

The authors must declare the protocol they have followed in the registration of anthropometric measurements (Body composition and morphological assessment of nutritional status in adults: a review of anthropometric variables. J Hum Nutr Diet. 2016 Feb;29(1):7-25 doi:10.1111/jhn.12278).

In this section it indicates that urine and blood biomarkers have been recorded, but in the introduction it does not indicate the importance of recording these biomarkers and the reason. The authors are reminded that, although they have recorded a multitude of data in this study, it does not mean that all of them are of interest for the stated objective.

This section indicates that the sample is divided according to the athlete's fluid intake (LD1, LD2, HD1 and HD2). However, this division of previous studies has not been justified or based on any theoretical argument based on knowledge published in impact journals.

In the results section

Based on the comments in the introduction and the methodology, the necessary changes should be made in this section.

In the discussion section

It must be decided whether the importance of the study is to analyze the relationship between hydration and the performance of athletes or their health. And each of the sections must be ordered according to the response of the general and/or specific objective.

In the conclusion section, it should be modified based on the changes indicated in the previous sections.

In the bibliography section

There are several references that are more than 10 years old, it is recommended to eliminate or replace the previous references from the year 2000 with others, unless their use is duly justified.

Some references are not in the format recommended by the journal and are in APA format.

Author Response

Authors' Response to Reviewers' Comments

Reviewer 2

Open Review

English language and style

( ) Extensive editing of English language and style required
( ) Moderate English changes required
( ) English language and style are fine/minor spell check required
(x) I don't feel qualified to judge about the English language and style

Yes

Can be improved

Must be improved

Not applicable

Does the introduction provide sufficient background and include all relevant references?

( )

( )

(x)

( )

Are all the cited references relevant to the research?

( )

( )

(x)

( )

Is the research design appropriate?

( )

( )

(x)

( )

Are the methods adequately described?

( )

(x)

( )

( )

Are the results clearly presented?

( )

( )

(x)

( )

Are the conclusions supported by the results?

( )

( )

(x)

( )

Comments and Suggestions for Authors

In the abstract in the abbreviation TWI it is pending to indicate its meaning. This section should be explained by itself, without the need to go into the full article. The recommendations for change indicated in the complete article should be applied.

Response: Thanks for your comments. It has been revised accordingly, that we added the explanation of TWI in the Abstract Section (Line 20, Page 1).

In the introduction section

In the first paragraph there is a sentence without a bibliography.

In the second paragraph, in line 64, I understand that it refers to reference 12, not to 11. In this same paragraph, in line 61, it indicates a study in an intake in liters and another in milliliters, it is recommended to use in both quantities the same unit of measure.

Response: Thanks for your comments. It had been revised accordingly.

We have added the references (references 15, 16; 17, 18) (Line 72-74, Page 2), and we revised the “1918 ml” into “19.18 L” (Line 70, Page 2).

References:

[1] Ma, G; Zhang, Q; Liu, A; Zuo, J; Zhang, W; Zou, S; Li, X; Lu, L; Pan, H; Hu, X. Fluid intake of adults in four Chinese cities. Nutr. Rev. 2012, 70 Suppl 2, S105-S110.

[2] Zhang Q, Hu X, Zhou S, Zuo J, Liu Z, Pan Q, Liu C, Pan H, Ma G (2011) Water intake of adults in four cities of China in summer. Chin J Prev Med 45:677–682. (in Chinese)

[3] Zhang, N.; Du, S.; Tang, Z.; Zheng, M.; Yan, R.; Zhu, Y.; Ma, G. Hydration, Fluid Intake, and Related Urine Biomarkers among Male College Students in Cangzhou, China: A Cross-Sectional Study-Applications for Assessing Fluid Intake and Adequate Water Intake. Int. J. Env. Res. Pub. He. 2017, 14, 513.

In the third paragraph the authors indicate the term free life, what do they mean by this term? It is recommended that the terms used are in the MeSH database (https://www.ncbi.nlm.nih.gov/mesh/) or that reviews and/or meta-analyses of this area of ​​knowledge have been previously published.

Response: Thanks for your comments.

The aims of the study were to investigate the drinking patterns and hydration biomarkers among young athletes in free-living conditions in China. The “free-living conditions” meant that we did not put any interventions on every aspects of their life, they could do everything as usual, including the class, the exercise, the eating and so on. They were only asked to record the fluids intake on the questionnaire each time they drank, and should take the fruits and snakes to the investigators to weight before and after they ate. In our previous study, the drinking patterns and hydration biomarkers were explored among free-living young males and females. Furthermore, subjects with higher total drinking fluids intake had better drinking patterns and hydration status than their counterparts with lower total drinking fluids intake. Moreover, there were many studies investigated the hydration status among free-living elderly volunteers, healthy adults, children, pregnant and lactating women. Therefore, we decided to use the item “free-living conditions” to mean "under normal living conditions".

Unfortunately, we did not search the Mesh Data for the “free-living” or the “free-living conditions” before. In the future, it may be better to search the related items in the MeSH database.

References:

[1] Zhang J, Zhang N, Wang Y, et al. Drinking patterns and hydration biomarkers among young adults with different levels of habitual total drinking fluids intake in Baoding, Hebei Province, China: a cross-sectional study[J]. BMC Public Health, 2020, 20:468.

[2] A Gonçalves, Silva J, Carvalho J, et al. Hydration status and water sources in free-living physically active elderly[J]. Nutricion Hospitalaria, 2015, 32(s02):10303.

[3] Bottin J H, Lemetais G, Poupin M, et al. Equivalence of afternoon spot and 24-h urinary hydration biomarkers in free-living healthy adults[J]. European journal of clinical nutrition, 2016, 70(8):904-7.

[4] Adams J D, Arnaoutis G, Johnson E C, et al. Combining urine color and void number to assess hydration in adults and children[J]. European Journal of Clinical Nutrition, 2021, 75(8):1262-1266.

[5] Heen E, Yassin A A, Madar A A, et al. Estimates of fluid intake, urine output and hydration-levels in women from Somaliland: a cross-sectional study[J]. Journal of nutrition science, 2021, 10:e66.

[6] Perrier E, Vergne S, Klein A, et al. Hydration biomarkers in free-living adults with different levels of habitual fluid consumption[J]. The British journal of nutrition, 2012, 109(9):1-10.

[7] Armstrong L E, Pumerantz A C, Fiala K A, et al. Human hydration indices: acute and longitudinal reference values[J]. International Journal of Sport Nutrition & Exercise Metabolism, 2010, 20(2):145-53.

[8] Armstrong L E, Johnson E C, Munoz C X, et al. Evaluation of Uosm:Posm ratio as a hydration biomarker in free-living, healthy young women[J]. European Journal of Clinical Nutrition, 2013, 67:934-938.

Several points of interest could be distinguished in this area of ​​knowledge: i. Water and hydration in sports, with the intake of carbohydrates and micronutrients to improve sports performance; ii. Methods for analyzing body weight loss after training or competition; iii. Changes in biochemical markers as an effect of hydration or dehydration. Despite these points described in the introduction, the objectives set are not justified. It is recommended to focus on the objective of analyzing the relationship in hydration in young athletes, for that perhaps the following bibliography can help:

  • Armstrong LE. Rehydration during Endurance Exercise: Challenges, Research, Options, Methods. Nutrients. 2021 Mar 9;13(3):887. doi: 10.3390/nu13030887
  • Baker LB, Ungaro CT, Sopeña BC, Nuccio RP, Reimel AJ, Carter JM, Stofan JR, Barnes KA. Body map of regional vs. whole body sweating rate and sweat electrolyte concentrations in men and women during moderate exercise-heat stress. J Appl Physiol (1985). 2018 May 1;124(5):1304-1318. doi: 10.1152/japplphysiol.00867.2017
  • Baker LB. Sweating Rate and Sweat Sodium Concentration in Athletes: A Review of Methodology and Intra/Interindividual Variability. Sports Med. 2017 Mar;47(Suppl 1):111-128. doi: 10.1007/s40279-017-0691-5
  • Barnes KA, Anderson ML, Stofan JR, Dalrymple KJ, Reimel AJ, Roberts TJ, Randell RK, Ungaro CT, Baker LB. Normative data for sweating rate, sweat sodium concentration, and sweat sodium loss in athletes: An update and analysis by sport. J Sports Sci. 2019 Oct;37(20):2356-2366. doi: 10.1080/02640414.2019.1633159
  • Campbell B, Wilborn C, La Bounty P, Taylor L, Nelson MT, Greenwood M, Ziegenfuss TN, Lopez HL, Hoffman JR, Stout JR, Schmitz S, Collins R, Kalman DS, Antonio J, Kreider RB. International Society of Sports Nutrition position stand: energy drinks. J Int Soc Sports Nutr. 2013 Jan 3;10(1):1. doi: 10.1186/1550-2783-10-1
  • Kenefick RW. Drinking Strategies: Planned Drinking Versus Drinking to Thirst. Sports Med. 2018 Mar;48(Suppl 1):31-37. doi: 10.1007/s40279-017-0844-6

If the target subjects are young adult athletes, it is recommended to delete all or most of the information on older adults and non-athletic adults. In addition, the authors must decide if this study is focused on improving the performance or health status of athletes.

Response: Thanks for your comments.

The objectives of the present study were to investigate the drinking patterns and hydration biomarkers among young athletes with different habitual total drinking fluids. In other words, the differences in the amounts, the types of the drinking fluids and the plasma and urinary biomarkers were evaluated among young athletes in our study.

In the Introduction Section, we wanted to illustrate the higher proportions of the insufficient intake of fluids among different ages of people, including the children and the elderly. Moreover, we have added more information of the fluids intake among athletes this time.

We have read and discussed the bibliography you recommended and we added them into the Introduction and Discussion Sections.

The objective of this study may be to analyze whether athletes apply good hydration guidelines. And in the event that they do not have good hydration guidelines, then in the discussion the recommendations that are necessary in the case of future studies will be indicated. Therefore, the comments on line 97 are not appropriate for this section.

In the methodology section

The authors must declare whether they have followed the recommendations of the Declaration of Helsinki (World Medical Association Declaration of Helsinki: ethical principles for medical research involving human subjects. JAMA. 2013 Nov 27;310(20):2191-4. doi: 10.1001/ jama.2013.281053) and inform if the study has been approved by an ethics committee.

Response: Thanks for your comments.

The study protocol was approved by the Peking University Institutional Review Committee. The ethical approval project identification code is IRB00001052-16071. This study was conducted according to the guidelines of the Declaration of Helsinki. All subjects signed an informed con-sent form before participating in the study.

All the information was put in the end of the manuscript (Lines 417-420, Page 10).

The type of sports practiced by the study subjects should be indicated and the scale used to indicate that the athletes performed moderate-intensity exercise should be indicated, as recommended by the scale proposed by the American College of Sports Medicine (Quantity and quality of exercise for developing and maintaining cardiorespiratory, musculoskeletal, and neuromotor fitness in apparently healthy adults: guidance for prescribing exercise. Med Sci Sports Exerc. 2011;43(7):1334-59.doi:10.1249/MSS.0b013e318213fefb).

Response: Thanks for your comments.

Participants were asked to recorded the information of the physical activity during the 7 days on the questionnaire, including the time, the types, the places of the activities, which was self-designed by the researchers. Furthermore, the training program of the participants were provided to the researchers by the coaches that guiding them. The types of sports of the athletes were aerobics, table tennis, tennis, football, track-and-field and others. There were no significant differences in the proportions of the types of sports among the athletes in the four groups (p>0.05), which had been added into the Methods Section (Lines 201-206, Page 5).

Furthermore, with reference to previous studies, considering the convenience in population studies and the accuracy of the results, we used an ActiGtraph WGTX3-BT triaxial accelerometer to monitor the physical activity of participants. But unfortunately, we only measured the activities of 45 of the 109 participants. All the 45 participants were required to wear the accelerometer continuously for 7 days, which cannot be removed except for swimming, bathing and sleeping. The monitors began to record data from the 0:00 of day 1 after issuance, and finished recording data at 24:00 on day 7, and was recovered on the eighth day.

According to the American College of Sports Medicine (ACSM) Position Stand “Quantity and quality of exercise for developing and maintaining cardiorespiratory, musculoskeletal, and neuromotor fitness in apparently healthy adults: guidance for prescribing exercise”, the MET of the 42 participants were 1.45±0.19, and the proportion of MVPA in the activities was 10.9±4.4 (unpublished data).

In the measurement of Total Water Intake, why are the methods described appropriate and not others?

Response: Thanks for your comments.

The total water intake was consisted of total drinking fluids and water from food. The total drinking fluids was assessed by a 7-day 24-h fluids questionnaire, which was an effective and reliable method of estimating total drinking fluids [1-4]. In our study, subjects were asked to record the time, volume, type and place for each fluid they consumed, which supplied the details necessary to explore the drinking patterns of the

subjects. In some studies, a food frequency questionnaire was used to assess the water from food or a 24-h recall was used to assess the total drinking fluids, which may underestimate the consumption of total drinking fluids as much as 500 mL/day [5]. In our study, all the foods were weighed before and after the subjects ate, and the samples of food were measured by professionals. Water from snacks including fruit was calculated according to the data from the Chinese food composition table (2009).

References:

[1] Ma G, Zhang Q, Liu A, Zuo J, Zhang W, Zou S, Li X, Lu L, Pan H, Hu X (2012) Fluid intake of adults in four Chinese cities. Nutr Rev 70: S105-S110.

[2] Du S, Hu X, Zhang Q, Wang X, Pan H, Gao J, Song J, Gao C, He Z, Ma G (2013) Water intake of primary and middle school students in four cities of China. Chin J Prev Med 3:210-213. (in Chinese)

[3] Johnson EC, Péronnet F, Jansen LT, Capitan-Jiménez C, Adams JD, Guelinckx I, Jiménez L, Mauromoustakos A, Kavouras SA (2017) Validation testing demonstrates efficacy of a 7-day fluid record to estimate daily water intake in adult men and women

when compared with total body water turnover measurement. J Nutr 147:2001-2007.

[4] Hernández-Cordero S, López-Olmedo N, Rodríguez-Ramírez S, Barquera-Cervera S, Rivera-Dommarco J, Popkin B (2015) Comparing a 7-day diary vs. 24 h-recall for estimating fluid consumption in overweight and obese Mexican women. BMC Public Health 15:1031.

[5] Vergne S (2012) Methodological aspects of fluid intake records and surveys. Nutr Today 47: S7-S10.

[6] Zhang N, Du S, Tang Z, et al. Hydration, Fluid Intake, and Related Urine Biomarkers among Male College Students in Cangzhou, China: A Cross-Sectional Study-Applications for Assessing Fluid Intake and Adequate Water Intake[J]. International Journal of Environmental Research and Public Health, 2017, 14(5).

[7] Zhang J, Zhang N, Wang Y, et al. Drinking patterns and hydration biomarkers among young adults with different levels of habitual total drinking fluids intake in Baoding, Hebei Province, China: a cross-sectional study[J]. BMC Public Health, 2020, 20:468.

The authors must declare the protocol they have followed in the registration of anthropometric measurements (Body composition and morphological assessment of nutritional status in adults: a review of anthropometric variables. J Hum Nutr Diet. 2016 Feb;29(1):7-25 doi:10.1111/jhn.12278).

Response: Thanks for your comments. We have revised accordingly, by adding the references into the manuscript (Lines 172-173, Page 4).

Height was measured twice to the nearest 0.1 cm, and weight was measured twice to the nearest 0.1 kg by trained investigators following standardized procedures with a height-weight meter (HDM-300; Huaju, Zhejiang, China), with the subjects wearing light clothing and no footwear.

We have followed the protocol that in the “Nutrition and food hygiene (Eighth Edition)” and the “Guidelines for the prevention and control of overweight and obesity in Chinese adults” of anthropometric measurements.

References:

[1] Nutrition and food hygiene (Eighth Edition). 2017, People's Health Publishing House.

[2] Guidelines for the prevention and control of overweight and obesity in Chinese adults. Acta Nutrimenta Sinica, 2004, 26:1-4.

In this section it indicates that urine and blood biomarkers have been recorded, but in the introduction it does not indicate the importance of recording these biomarkers and the reason. The authors are reminded that, although they have recorded a multitude of data in this study, it does not mean that all of them are of interest for the stated objective.

Response: Thanks for your comments.

In the Introduction Section, we declared that the dinking patterns may impact the hydration status of the athletes with different habitual total drinking fluids. However, we did not the importance of the hydration status for the athletes and the types of the hydration biomarkers. Therefore, we added them into the Introduction Section (Lines 96-110, Pages 2-3).

This section indicates that the sample is divided according to the athlete's fluid intake (LD1, LD2, HD1 and HD2). However, this division of previous studies has not been justified or based on any theoretical argument based on knowledge published in impact journals.

Response: Thanks for your comments.

In our previous study, young adults with different levels of habitual total drinking fluids had different drinking patterns and hydration biomarkers in China. They were divided into four groups, according to the quartile of the total drinking fluids. Then, in the present study, we wanted to know if the athletes with different total drinking fluids had the same changes of drinking patterns and hydration biomarkers, as the young adults in our previous study. Then, we also divided the subjects into four groups.

Among the young adults in France, it also confirmed the conclusion that, young adults including males and female with low consumption of water (≤1.2 L/d) had more concentrated urine, and higher concentrations of cortisol, creatinine and arginine vasopressin than those with high consumption (>2.0 L/d).

References:

[1] Zhang J, Zhang N, Wang Y, et al. Drinking patterns and hydration biomarkers among young adults with different levels of habitual total drinking fluids intake in Baoding, Hebei Province, China: a cross-sectional study[J]. BMC Public Health, 2020, 20.

[2] Perrier E, Vergne S, Klein A, et al. Hydration biomarkers in free-living adults with different levels of habitual fluid consumption[J]. The British journal of nutrition, 2012, 109(9):1-10.

[3] Ferreira-Pego C, Guelin Ck X I, Moreno L A, et al. Total fluid intake and its determinants: cross-sectional surveys among adults in 13 countries worldwide. European Journal of Nutrition, 2015, 54(2):S35-S43.u

In the results section

Based on the comments in the introduction and the methodology, the necessary changes should be made in this section.

Response: Thanks for your comments. We have revised accordingly.

In the discussion section

It must be decided whether the importance of the study is to analyze the relationship between hydration and the performance of athletes or their health. And each of the sections must be ordered according to the response of the general and/or specific objective.

In the conclusion section, it should be modified based on the changes indicated in the previous sections.

Response: Thanks for your comments.

The objectives of the present study were to explore the differences of the drinking patterns and the hydration status among athletes with different habitual total drinking fluids intake. The drinking patterns included the amounts, and the types of the fluids intake. The hydration status was measured by the plasma and urinary biomarkers.

It has been revised accordingly in the Discussion Section, based on the changes of the Introduction and Results Sections.

In the bibliography section

There are several references that are more than 10 years old, it is recommended to eliminate or replace the previous references from the year 2000 with others, unless their use is duly justified.

Response: Thanks for your comments.

We added some new references that related to our study (Lines 436-563, Pages 11-12).

Some references are not in the format recommended by the journal and are in APA format.

Response: Thanks for your comments.

We have revised the format of the references recommended by the journal (Lines 436-563, Pages 11-12).
